# Partially Encrypted Machine Learning using Functional Encryption

Théo Ryffel[1,2], Edouard Dufour-Sans[1], Romain Gay[1,3],
Francis Bach[2,1] and David Pointcheval[1,2]

[1]Département d'informatique de l'ENS, ENS, CNRS, PSL University, Paris, France
[2]INRIA, Paris, France
[3]University of California, Berkeley

{theo.ryffel,edufoursans,romain.gay,francis.bach,david.pointcheval}@ens.fr

## Abstract

Machine learning on encrypted data has received a lot of attention thanks to recent breakthroughs in homomorphic encryption and secure multi-party computation. It allows outsourcing computation to untrusted servers without sacrificing privacy of sensitive data. We propose a practical framework to perform partially encrypted and privacy-preserving predictions which combines adversarial training and functional encryption. We first present a new functional encryption scheme to efficiently compute quadratic functions so that the data owner controls what can be computed but is not involved in the calculation: it provides a decryption key which allows one to learn a specific function evaluation of some encrypted data. We then show how to use it in machine learning to partially encrypt neural networks with quadratic activation functions at evaluation time, and we provide a thorough analysis of the information leaks based on indistinguishability of data items of the same label. Last, since most encryption schemes cannot deal with the last thresholding operation used for classification, we propose a training method to prevent selected sensitive features from leaking, which adversarially optimizes the network against an adversary trying to identify these features. This is interesting for several existing works using partially encrypted machine learning as it comes with little reduction on the model's accuracy and significantly improves data privacy.

## 1 Introduction

As both public opinion and regulators are becoming increasingly aware of issues of data privacy, the area of privacy-preserving machine learning has emerged with the aim of reshaping the way machine learning deals with private data. Breakthroughs in fully homomorphic encryption (FHE) [15, 18] and secure multi-party computation (SMPC) [19, 39] have made computation on encrypted data practical and implementations of neural networks to do encrypted predictions have flourished [34–36, 8, 13].

However, these protocols require the data owner encrypting the inputs and the parties performing the computations to interact and communicate in order to get decrypted results, which we would like to avoid in some cases, like spam filtering, for example, where the email receiver should not need to be online for the email server to classify incoming email as spam or not. Functional encryption (FE) [12, 32] in return does not need interaction to compute over encrypted data: it allows users to receive in plaintext specific functional evaluations of encrypted data: for a function $f$, a functional decryption key can be generated such that, given any ciphertext with underlying plaintext $x$, a user can use this key to obtain $f(x)$ without learning $x$ or any other information than $f(x)$. It stands in

between traditional public key encryption, where data can only be directly revealed, and FHE, where data can be manipulated but cannot be revealed: it allows the user to tightly control what is disclosed about his data.

## 1.1 Use cases

**Spam filtering.** Consider the following scenario: Alice uses a secure email protocol which makes use of functional encryption. Bob uses Alice's public key to send her an email, which lands on Alice's email provider's server. Alice gave the server keys that enable it to process the email and take a predefined set of appropriate actions without her being online. The server could learn how urgent the email is and decide accordingly whether to alert Alice. It could also detect whether the message is spam and store it in the spam box right away.

**Privacy-preserving enforcement of content policies** Another use case could be to enable platforms, such as messaging apps, to maintain user privacy through end-to-end encryption, while filtering out content that is illegal or doesn't adhere to policies the site may have regarding, for instance, abusive speech or explicit images.

These applications are not currently feasible within a reasonable computing time, as the construction of FE for all kinds of circuits is essentially equivalent to *indistinguishable obfuscation* [7, 21], concrete instances of which have been shown insecure, let alone efficient. However, there exist practical FE schemes for the inner-product functionality [1, 2] and more recently for quadratic computations [6], that is usable for practical applications.

## 1.2 Our contributions

We introduce a new FE scheme to compute quadratic forms which outperforms that of Baltico et al. [6] in terms of complexity, and provide an efficient implementation of this scheme. We show how to use it to build privacy preserving neural networks, which perform well on simple image classification problems. Specifically, we show that the first layers of a polynomial network can be run on encrypted inputs using this quadratic scheme.

In addition, we present an adversarial training technique to process these first layers to improve privacy, so that their output, which is in plaintext, cannot be used by adversaries to recover specific sensitive information at test time. This adversarial procedure is generic for semi-encrypted neural networks and aims at reducing the information leakage, as the decrypted output is not directly the classification result but an intermediate layer (i.e., the neuron outputs of the neural network before thresholding). This has been overlooked in other popular encrypted classification schemes (even in FHE-based constructions like [20] and [15]), where the argmax operation used to select the class label is made in clear, as it is either not possible with FE, or quite inefficient with FHE and SMPC.

We demonstrate the practicality of our approach using a dataset inspired from MNIST [27], which is made of images of digits written using two different fonts. We show how to perform classification of the encrypted digit images in less than 3 seconds with over 97.7% accuracy while making the font prediction a hard task for a whole set of adversaries.

This paper builds on a preliminary version available on the Cryptology ePrint Archive at `eprint.iacr.org/2018/206`. All code and implementations can be found online at `github.com/LaRiffle/collateral-learning` and `github.com/edufoursans/reading-in-the-dark`.

## 2 Background Knowledge

### 2.1 Quadratic and Polynomial Neural Networks

Polynomial neural networks are a class of networks which only use linear elements, like fully connected linear layers, convolutions but with average pooling, and model activation functions with polynomial approximations when not simply the square function. Despite these simplifications, they have proved themselves satisfactorily accurate for relatively simple tasks ([20] learns on MNIST and [5] on CIFAR10 [26]). The simplicity of the operations they build on guarantees good efficiency, especially for the gradient computations, and works like [28] have shown that they can achieve convergence rates similar to those of networks with non-linear activations.

In particular, they have been used for several early stage implementations in cryptography [20, 18, 14] to demonstrate the usability of new protocols for machine learning. However, the argmax or other thresholding function present at the end of a classifier network to select the class among the output neurons cannot be conveniently handled, so several protocol implementations (among which ours) run polynomial networks on encrypted inputs, but take the argmax over the decrypted output of the network. This results in potential information leakage which could be maliciously exploited.

## 2.2 Functional Encryption

Functional encryption extends the notion of public key encryption where one uses a public key pk and a secret key sk to respectively encrypt and decrypt some data. More precisely, pk is still used to encrypt data, but for a given function $f$, sk can be used to derive a functional decryption key $\mathsf{dk}_f$ which will be shared to users so that, given a ciphertext of $x$, they can decrypt $f(x)$ but not $x$. In particular, someone having access to $\mathsf{dk}_f$ cannot learn anything about $x$ other than $f(x)$. Note also that functions cannot be composed, since the decryption happens within the function evaluation. Hence, only single quadratic functions can be securely evaluated. A formal definition of functional encryption is provided in Appendix A.1.

**Perfect correctness.** Perfect correctness is achieved in functional encryption: $\forall x \in \mathcal{X}$, $f \in \mathcal{F}$, $\Pr[\mathsf{Dec}(\mathsf{dk}_f, \mathsf{ct}) = f(x)] = 1$, where $\mathsf{dk}_f \leftarrow \mathsf{KeyGen}(\mathsf{msk}, f)$ and $\mathsf{ct} \leftarrow \mathsf{Enc}(\mathsf{pk}, x)$. Note that this property is a very strict condition, which is not satisfied by exisiting fully homomorphic encryption schemes (FHE), such as [16, 22].

## 2.3 Indistinguishability and security

To assess the security of our framework, we first consider the FE scheme security and make sure that we cannot learn anything more than what the function is supposed to output given an encryption of $x$. Second, we analyze how sensitive the output $f(x)$ is with respect to the private input $x$. For both studies, we will rely on *indistinguishability* [23], a classical security notion which can be summed up in the following game: an adversary provides two input items to the challenger (here our FE algorithm), and the challenger chooses one item to be encrypted, runs encryption on it before returning the output. The adversary should not be able to detect which input was used. This is known as IND-CPA security in cryptography and a formal definition of it can be found in Appendix A.2.

We will first prove that our quadratic FE scheme achieves IND-CPA security, then, we will use a relaxed version of indistinguishability to measure the FE output sensitivity. More precisely, we will make the hypothesis that our input data can be used to predict public labels but also sensitive private ones, respectively $y_{\mathrm{pub}}$ and $y_{\mathrm{priv}}$. Our quadratic FE scheme $q$ aims at predicting $y_{\mathrm{pub}}$ and an adversary would rather like to infer $y_{\mathrm{priv}}$. In this case, the security game consists in the adversary providing two inputs $(x_0, x_1)$ labelled with the same $y_{\mathrm{pub}}$ but a different $y_{\mathrm{priv}}$ and then trying to distinguish which one was selected by the challenger, given its output $q(x_b)$, $b \in \{0, 1\}$. One way to do this is to measure the ability of an adversary to predict $y_{\mathrm{priv}}$ for items which all belong to the same $y_{\mathrm{pub}}$ class.

In particular, note that we do not consider approaches based on input reconstruction (as done by [17]) because in many cases, the adversary is not interested in reconstructing the whole input, but rather wants to get insights into specific characteristics.

Another way to see this problem is that we want the sensitive label $y_{\mathrm{priv}}$ to be independent from the decrypted output $q(x)$ (which is a proxy to the prediction), given the true public label $y_{\mathrm{pub}}$. This independence notion is known as *separation* and is used as a fairness criterion in [9] if the sensitive features can be misused for discrimination.

# 3 Our Context for Private Inference

## 3.1 Classifying in two directions

We are interested in specific types of datasets $(\vec{x}_i)_{i=1,\dots,n}$ which have public labels $y_{\mathrm{pub}}$ but also private ones $y_{\mathrm{priv}}$. Moreover, these different types of labels should be entangled, meaning that they should not be easily separable, unlike the color and the shape of an object in an image for example which can be simply separated. For example, in the spam filtering use case mentioned above, $y_{\mathrm{pub}}$ would be a spam flag, and $y_{\mathrm{priv}}$ would be some marketing information highlighting areas of interest

of the email recipient like technology, culture, etc. In addition, to simplify our analysis, we assume that classes are balanced for all types of labels, and that labels are independent from each other given the input: $\forall \vec{x}, P(y_{\text{pub}}, y_{\text{priv}}|\vec{x}) = P(y_{\text{pub}}|\vec{x})P(y_{\text{priv}}|\vec{x})$. To illustrate our approach in the case of image recognition, we propose a synthetic dataset inspired by MNIST which consists of 60 000 grey scaled images of $28 \times 28$ pixels representing digits using two fonts and some distortion, as shown in Figure 1. Here, the public label $y_{\text{pub}}$ is the digit on the image and the private one $y_{\text{priv}}$ is the font used to draw it.

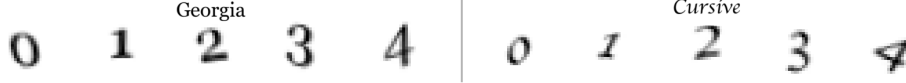

Figure 1: Artificial dataset inspired from MNIST with two types of labels.

We define two tasks: a *main* task which tries to predict $y_{\text{pub}}$ using a partially-encrypted polynomial neural network with functional encryption, and a *collateral* task which is performed by an adversary who tries to leverage the output of the FE encrypted network at test time to predict $y_{\text{priv}}$. Our goal is to perform the main task with high accuracy while making the collateral one as bad as random predictions. In terms of indistinguishability, given a dataset with the same digit drawn, it should be infeasible to detect the used font.

## 3.2 Equivalence with a Quadratic Functional Encryption scheme

We now introduce our new framework for quadratic functional encryption and show that it can be used to partially encrypt a polynomial network.

### 3.2.1 Functional Encryption for Quadratic Polynomials

We build an efficient FE scheme for the set of quadratic functions defined as $\mathcal{F}_{n,B_x,B_y,B_q} \subset \{q : [-B_x, B_x]^n \times [-B_y, B_y]^n \rightarrow \mathbb{Z}\}$, where $q$ is described as a set of bounded coefficients $\{q_{i,j} \in [-B_q, B_q]\}_{i,j \in [n]}$ and for all vectors $(\vec{x}, \vec{y})$, we have $q(\vec{x}, \vec{y}) = \sum_{i,j \in [n]} q_{i,j} x_i y_j$.

A complete version of our scheme is given in Figure 2, but here are the main ideas and notations. First note that we use bilinear groups, *i.e.*, a set of prime-order groups $\mathbb{G}_1$, $\mathbb{G}_2$ and $\mathbb{G}_T$ together with a bilinear map $e : \mathbb{G}_1 \times \mathbb{G}_2 \rightarrow \mathbb{G}_T$ called *pairing* which satisfies $e(g_1^a, g_2^b) = e(g_1, g_2)^{ab}$ for any exponents $a, b \in \mathbb{Z}$: one can compute quadratic polynomials in the exponent. Here, $g_1$, $g_2$ are generators of $\mathbb{G}_1$ and $\mathbb{G}_2$ and $g_T := e(g_1, g_2)$ is a generator of the target group $\mathbb{G}_T$. A pair of vectors $(\vec{s}, \vec{t})$ is first selected and constitutes the private key msk, while the public key is $(g_1^{\vec{s}}, g_2^{\vec{t}})$.

Encrypting $(\vec{x}, \vec{y})$ roughly consists of masking $g_1^{\vec{x}}$ and $g_2^{\vec{y}}$ with $g_1^{\vec{s}}$ and $g_2^{\vec{t}}$, which allows any user to compute $g_T^{q(\vec{x},\vec{y})-q(\vec{s},\vec{t})}$ with for any quadratic function $q$, using the pairing. The functional decryption key for a specific $q$ is $g_T^{q(\vec{s},\vec{t})}$ which allows to get $g_T^{q(\vec{x},\vec{y})}$. Last, taking the discrete logarithm gives access to $q(\vec{x}, \vec{y})$ (discrete logarithm for small exponents is easy). Security uses the fact that it is hard to compute msk from pk (discrete logarithm for large exponents $\vec{s}, \vec{t}$ is hard to compute). More details are given in Appendix B.1[1]

**Theorem 3.1 (Security, correctness and complexity)** *The FE scheme provided in Figure 2:*

- *is IND-CPA secure in the Generic Bilinear Group Model,*
- *verifies* $\log(out) = q(\vec{x}, \vec{y})$ *and satisfies perfect correctness,*
- *has a overall decryption complexity of* $2n^2(E + P) + P + D,$

*where $E$, $P$ and $D$ respectively denote exponentiation, pairing and discrete logarithm complexities.*

Our scheme outperforms previous schemes for quadratic FE with the same security assumption, like the one from [6, Sec. 4] which achieves $3n^2(E+P)+2P+D$ complexity and uses larger ciphertexts and decryption keys. Note that the efficiency of the decryption can even be further optimized for those quadratic polynomials used that are relevant to our application (see Section 3.2.2).

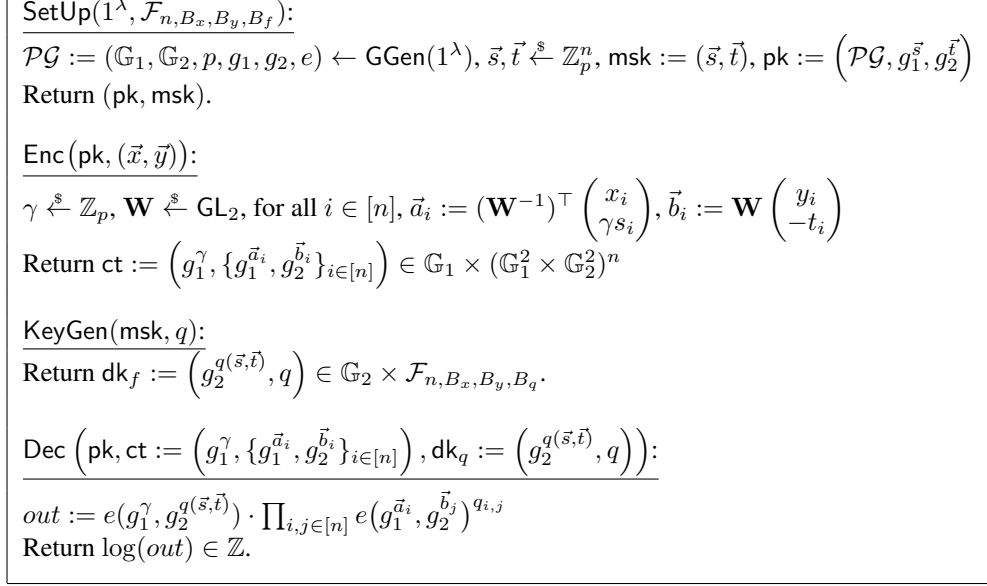

$\underline{\mathsf{SetUp}(1^\lambda, \mathcal{F}_{n,B_x,B_y,B_f})\text{:}}$

$\mathcal{PG} := (\mathbb{G}_1, \mathbb{G}_2, p, g_1, g_2, e) \leftarrow \mathsf{GGen}(1^\lambda),\ \vec{s}, \vec{t} \overset{\$}{\leftarrow} \mathbb{Z}_p^n,\ \mathsf{msk} := (\vec{s}, \vec{t}),\ \mathsf{pk} := \left( \mathcal{PG}, g_1^{\vec{s}}, g_2^{\vec{t}} \right)$

Return $(\mathsf{pk}, \mathsf{msk})$.

$\underline{\mathsf{Enc}\big(\mathsf{pk}, (\vec{x}, \vec{y})\big)\text{:}}$

$\gamma \overset{\$}{\leftarrow} \mathbb{Z}_p,\ \mathbf{W} \overset{\$}{\leftarrow} \mathsf{GL}_2,$ for all $i \in [n],\ \vec{a}_i := (\mathbf{W}^{-1})^\top \begin{pmatrix} x_i \\ \gamma s_i \end{pmatrix},\ \vec{b}_i := \mathbf{W} \begin{pmatrix} y_i \\ -t_i \end{pmatrix}$

Return $\mathsf{ct} := \left( g_1^\gamma, \{g_1^{\vec{a}_i}, g_2^{\vec{b}_i}\}_{i \in [n]} \right) \in \mathbb{G}_1 \times (\mathbb{G}_1^2 \times \mathbb{G}_2^2)^n$

$\underline{\mathsf{KeyGen}(\mathsf{msk}, q)\text{:}}$

Return $\mathsf{dk}_f := \left( g_2^{q(\vec{s},\vec{t})}, q \right) \in \mathbb{G}_2 \times \mathcal{F}_{n,B_x,B_y,B_q}.$

$\underline{\mathsf{Dec}\left( \mathsf{pk}, \mathsf{ct} := \left( g_1^\gamma, \{g_1^{\vec{a}_i}, g_2^{\vec{b}_i}\}_{i \in [n]} \right), \mathsf{dk}_q := \left( g_2^{q(\vec{s},\vec{t})}, q \right) \right)\text{:}}$

$out := e(g_1^\gamma, g_2^{q(\vec{s},\vec{t})}) \cdot \prod_{i,j \in [n]} e\big(g_1^{\vec{a}_i}, g_2^{\vec{b}_j}\big)^{q_{i,j}}$

Return $\log(out) \in \mathbb{Z}.$

Figure 2: Our functional encryption scheme for quadratic polynomials.

**Computing the discrete logarithm for decryption.** Our decryption requires computing discrete logarithms of group elements in base $g_T$, but contrary to previous works like [25] it is independent of the ciphertext and the functional decryption key used to decrypt. This allows to pre-compute values and dramatically speeds-up decryption.

### 3.2.2 Equivalence of the FE scheme with a Quadratic Network

We classify data which can be represented as a vector $\vec{x} \in [0, B]^n$ (in our case, the size $B = 255$, and the dimension $n = 784$) and we first build models $(q_i)_{i \in [\ell]}$ for each public label $i \in [\ell]$, such that our prediction $y_{\mathrm{pub}}$ for $\vec{x}$ is $\operatorname{argmax}_{i \in [\ell]} q_i(\vec{x})$.

**Quadratic polynomial on $\mathbb{R}^n$.** The most straightforward way to use our FE scheme would be for us to learn a model $(\mathbf{Q}_i)_{i \in [\ell]} \in (\mathbb{R}^{n \times n})^\ell$, which we would then round onto integers, such that $q_i(\vec{x}) = \vec{x}^\top \mathbf{Q}_i \vec{x}, \forall i \in [\ell]$. This is a unnecessarily powerful model in the case of MNIST as it has $\ell n^2$ parameters ($n = 784$), and the resulting number of pairings to compute would be unreasonably large.

**Linear homomorphism.** The encryption algorithm of our FE scheme is linearly homomorphic with respect to the plaintext: given an encryption of $(\vec{x}, \vec{y})$ under the secret key $\mathsf{msk} := (\vec{s}, \vec{t})$, one can efficiently compute an encryption of $(\vec{u}^\top \vec{x}, \vec{v}^\top \vec{y})$ under the secret key $\mathsf{msk}' := (\vec{u}^\top \vec{s}, \vec{v}^\top \vec{t})$ for any linear combination $\vec{u}, \vec{v}$ (see proof in Appendix B.2). Any vector $\vec{v}$ is a column, and $\vec{v}^\top$ is a row.

Therefore, if $q$ can be written $q(\vec{x}, \vec{y}) = (\mathbf{U}\vec{x})^\top \mathbf{M}(\mathbf{V}\vec{y})$ for all $(\vec{x}, \vec{y})$, with $\mathbf{U}, \mathbf{V} \in \mathbb{Z}_p^{d \times n}$ projection matrices and $\mathbf{M} \in \mathbb{Z}_p^{d \times d}$, it is more efficient to first compute the encryption of $(\mathbf{U}\vec{x}, \mathbf{V}\vec{y})$ from the encryption of $(\vec{x}, \vec{y})$, and then to apply the functional decryption on these ciphertexts, because their underlying plaintexts are of reduced dimension $d < n$. This reduces the number of exponentiations from $2n^2$ to $2dn$ and the number of pairing computations from $2n^2$ to $2d^2$ for a single $q_i$. This is a major efficiency improvement for small $d$, as pairings are the main bottleneck in the computation.

**Projection and quadratic polynomial on $\mathbb{R}^d$.** We can use this and apply the quadratic polynomials on projected vectors: we learn $\mathbf{P} \in \mathbb{R}^{n \times d}$ and $(\mathbf{Q}_i)_{i \in [\ell]} \in \left( \mathbb{R}^{d \times d} \right)^\ell$, and our model is $q_i(\vec{x}) = (\mathbf{P}\vec{x})^\top \mathbf{Q}_i(\mathbf{P}\vec{x}), \forall i \in [\ell]$. We only need $2\ell d^2$ pairings and since the same $\mathbf{P}$ is used for all $q_i$, we only compute once the encryption of $\mathbf{P}\vec{x}$ from the encryption of $\vec{x}$. Better yet, we can also perform the pairings only once, and then compute the scores by exponentiating with different coefficients the same results of the pairings, thus only requiring $2d^2$ pairing evaluations, independently of $\ell$.

**Degree 2 polynomial network, with one hidden layer.** To further reduce the number of pairings, we actually limit ourselves to diagonal matrices, and thus rename $\mathbf{Q}_i$ to $\mathbf{D}_i$. We find that the gain in efficiency associated with only computing $2d$ pairings is worth the small drop in accuracy. The resulting model is actually a polynomial network of degree 2 with one hidden layer of $d$ neurons and the activation function is the square. In the following experiments we take $d = 40$.

Our final encrypted model can thus be written as $q_i(\vec{x}) = (\mathbf{P}\vec{x})^\top \mathbf{D}_i(\mathbf{P}\vec{x}), \forall i \in [\ell]$, where we add a bias term to $\vec{x}$ by replacing it with $\vec{x} = (1 \ x_1 \ldots x_n)$.

**Full network.** The result of the quadratic $(q_i(\vec{x}))_{i \in [\ell]}$ (i.e., of the private quadratic network) is now visible in clear. As mentioned above, we cannot compose this block several times as it contains decryption, so this is currently the best that we can have as an encrypted computation with FE. Instead of simply applying the argmax to the cleartext output of this privately-evaluated quadratic network to get the label, we observe that adding more plaintext layers on top of it helps improving the overall accuracy of the main task. We have therefore a neural network composed of a private and a public part, as illustrated in Figure 3.

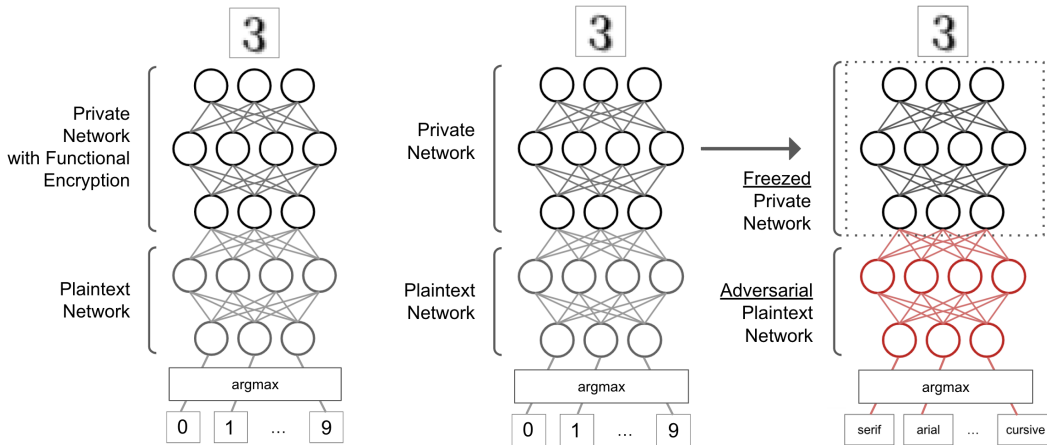

Figure 3: Semi-encrypted network using quadratic FE.

Figure 4: Semi-encrypted network with an adversary trying to recover private labels from the private quadratic network.

### 3.3 Threat of Collateral Learning

A typical adversary would have a read access to the main task classification process. It would leverage the output of the quadratic network to try to learn the font used on ciphered images. To do this, all that is needed is to train another network on top of the quadratic network so that it learns to predict the font, assuming some access to labeled samples (which is the case if the adversary encrypts itself images and provides them to the main task at evaluation time). Note that in this case the private network is not updated by the collateral network as we assume it is only provided in read access after the main task is trained. Figure 4 summarizes the setting.

We implemented this scenario using as adversary a neural network composed of a first layer acting as a decompression step where we increase the number of neurons from 10 back to $28 \times 28$ and add on top of it a classical[2] convolutional neural network (CNN). This structure is reminiscent of autoencoders [38] where the bottleneck is the public output of the private net and the challenge of this autoencoder is to correctly memorize the public label while forgetting the private one. What we observed is striking: in less than 10 epochs, the collateral network leverages the 10 public neurons output and achieves $93.5\%$ accuracy for the font prediction. As expected, it gets even worse when the adversary is assessed with the indistinguishability criterion because in that case the adversary can work on a dataset where only a specific digit is represented: this reduces the variability of the samples and makes it easier to distinguish the font; the probability of success is indeed of $96.9\%$.

We call *collateral learning* this phenomenon of learning unexpected features and will show in the next section how to implement counter-measures to this threat in order to improve privacy.

## 4 Defeating Collateral Learning

### 4.1 Reducing information leakage

Our first approach is based on the observation that we leak many bits of information. We first investigate whether we can reduce the number of outputs of the privately-evaluated network, as adding extra layers on top of the private network makes it no longer necessary to keep 10 of them.

The intuition is that if the information that is relevant to the main task can fit in less than 10 neurons, then the extra neurons would leak unnecessary information. We have therefore a trade-off between reducing too much and losing desired information or keeping a too large output and having an important leakage. We can observe this through the respective accuracies as it is shown in Figure 5, where the main and adversarial networks are CNNs as in Section 3.3 with 10 epochs of training using 7-fold cross validation. What we observe here is interesting: the main task does not exhibit significant weaknesses even with size 3 where we drop to $97.1\%$ which is still very good although $2\%$ under the best accuracy. In return, the collateral accuracy starts to significantly decrease when output size is below 7. At size 4, it is only $76.4\%$ on average so $18\%$ less than the baseline. We will keep an output size of 3 or 4 for the next experiments to keep the main accuracy almost unchanged.

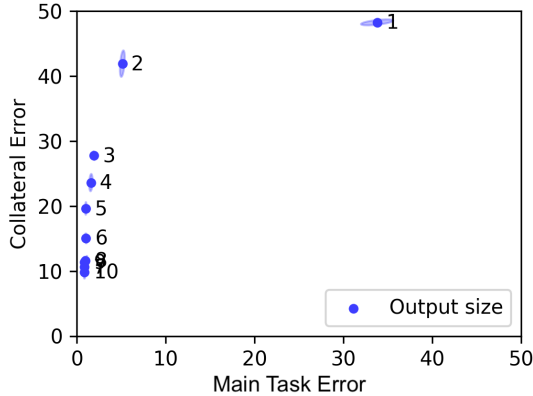

Figure 5: Trade-off between main and collateral accuracies depending on the private output size.

Another hyperparameter that we can consider is the weight compression: how many bits do we need to represent the weights on the private networks layers? This is of interest for the FE scheme as we need to convert all weights to integers and those integers will be low provided that the compression rate is high. Small weight integers mean that the output of the private network has a relatively low amplitude and can be therefore efficiently decrypted using discrete logarithm. We managed to express all weights and even the input image using 4 bit values with limited impact on the main accuracy and almost none on the collateral one. Details about compression can be found in Appendix C.1.

### 4.2 Adversarial training

We propose a new approach to actively adapt against collateral learning. The main idea is to simulate adversaries and to try to defeat them. To do this, we use semi-adversarial training and optimize simultaneously the main classification objective and the opposite of the collateral objective of a given simulated adversary. The function that we want to minimize at each iteration step can be written:

$$\min_{\theta_q}[\min_{\theta_{\text{pub}}} \mathcal{L}_{\text{pub}}(\theta_q, \theta_{\text{pub}}) - \alpha \min_{\theta_{\text{pub}}} \mathcal{L}_{\text{priv}}(\theta_q, \theta_{\text{pub}})].$$

This approach is inspired from [29] where the authors train some objective against nuisances parameters to build a classifier independent from these nuisances. Private features leaking in our scheme can indeed be considered to be a nuisance. However, our approach goes one step further as we do not just stack a network above another; our global network structure is fork-like: the common basis is the private network and the two forks are the main and collateral classifiers. This allows us to have a better classifier for the main task which is not as sensitive to the adversarial training as the scheme exposed by [29, Figure 1]. One other difference is that the collateral classifier is a specific modeling of an adversary, and we will discuss this in details in the next section. We define in Figure 6 the 3-step procedure used to implement this semi-adversarial training using partial back-propagation.

## 5 Experimental Results

**Accurate main task and poor collateral results.** In Figures 7 and 8 we show that the output size has an important influence on the two tasks' performances. For this experiment, we use $\alpha = 1.7$ as detailed in Appendix C.2, the adversary uses the same CNN as stated above and the main network is

Pre-training: *Initial phase where both tasks learn and strengthen before the joint optimization*
$\overline{\text{Minimize } \mathcal{L}_{\text{pub}}(\theta_q, \theta_{\text{pub}})}$
Minimize $\mathcal{L}_{\text{priv}}(\text{Frozen}(\theta_q), \theta_{\text{priv}})$

Semi-adversarial training: *The joint optimization phase, where $\theta_{\text{pub}}$ and $\theta_{\text{priv}}$ are updated depending on*
$\overline{\text{the variations of } \theta_q \text{ and } \theta_q \text{ is optimized to reduce the loss } \mathcal{L} = \mathcal{L}_{\text{pub}} - \alpha \mathcal{L}_{\text{priv}}}$
Minimize $\mathcal{L}_{\text{pub}}(\text{Frozen}(\theta_q), \theta_{\text{pub}})$
Minimize $\mathcal{L}_{\text{priv}}(\text{Frozen}(\theta_q), \theta_{\text{priv}})$
Minimize $\mathcal{L} = \mathcal{L}_{\text{pub}}(\theta_q, \text{Frozen}(\theta_{\text{pub}})) - \alpha \mathcal{L}_{\text{priv}}(\theta_q, \text{Frozen}(\theta_{\text{pub}}))$

Recover phase: *Both tasks recover from the perturbations induced by the adversarial phase, $\theta_q$ does not*
$\overline{\textit{change anymore}}$
Minimize $\mathcal{L}_{\text{pub}}(\text{Frozen}(\theta_q), \theta_{\text{pub}})$
Minimize $\mathcal{L}_{\text{priv}}(\text{Frozen}(\theta_q), \theta_{\text{priv}})$

Figure 6: Our semi-adversarial training scheme.

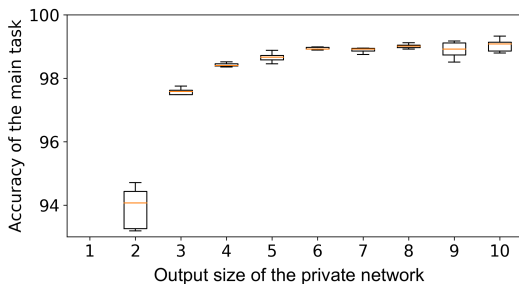

Figure 7: Influence of the output size on the main task accuracy with adversarial training.

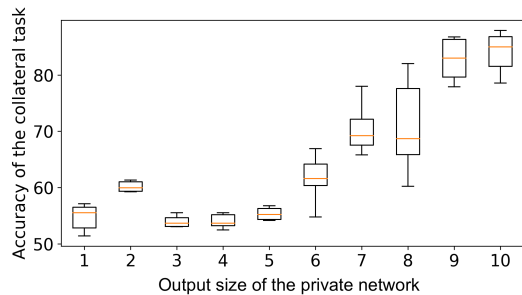

Figure 8: Influence of the output size on the collateral task accuracy with adversarial training.

a simple feed forward network (FFN) with 4 layers. We observe that both networks behave better when the output size increases, but the improvement is not synchronous which makes it possible to have a main task with high accuracy while the collateral task is still very inaccurate. In our example, this corresponds to an output size between 3 and 5. Note that the collateral result is the accuracy at the distinction task, i.e., the digit is fixed for the adversary which trains to distinguish two fonts during a 50 epoch *recover phase* using 7-fold cross validation, after 50 epochs of *semi-adversarial training* have been spent to reduce leakage from the private network.

**Generalizing resistance against multiple adversaries.** In practice, it is very likely that the adversary will use a different model than the one against which the protection has been built. We have therefore investigated how building resistance against a model $M$ can provide resistance against other models. Our empirical results tend to show that models with less parameters than $M$ do not perform well. In return, models with more parameters can behave better, provided that the complexity does not get excessive for the considered task, because it would not provide any additional advantage and would just lead to learning noise. In particular, the CNN already mentioned above seems to be a sufficiently complex model to resist against a wide range of feed forward (FFN) and convolutional networks, as illustrated in Figure 9 where the measure used is indistinguishability of the font for a fixed digit. This study is not exhaustive as the adversary can change the activation function (here we use relu) or even the training parameters (optimizer, batch size, dropout, etc.), but these do not seem to provide any decisive advantage.

We also assessed the resistance to a large range of other models from the sklearn library [33] and report the collateral accuracy in Figure 10. As can be observed, some models such as k-nearest neighbors or random forests perform better compared to neural networks, even if their accuracy remains relatively low. One reason can be that they operate in a very different manner compared to the model on which the adversarial training is performed: k-nearest neighbors for example just considers distances between points.

**Runtime.** Training in semi-adversarial mode can take quite a long time depending on the level of privacy one wants to achieve. However, the runtime during the test phase is much faster, it is dominated by the FE scheme part which can be broken down to 4 steps: functional key generation,

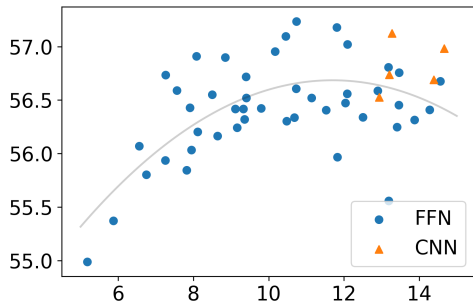

Figure 9: Collateral accuracy depending of the adversarial network complexity seen as the log of the number of parameters.

| | |
|---|---|
| Linear Ridge Regression | $53.5 \pm 0.5\%$ |
| Logistic Regression | $52.5 \pm 0.6\%$ |
| Quad. Discriminant Analysis | $54.9 \pm 0.3\%$ |
| SVM (RBF kernel) | $57.9 \pm 0.4\%$ |
| Gaussian Process Classifier | $53.8 \pm 0.3\%$ |
| Gaussian Naive Bayes | $53.2 \pm 0.5\%$ |
| K-Neighbors Classifier | $58.1 \pm 0.7\%$ |
| Decision Tree Classifier | $56.8 \pm 0.4\%$ |
| Random Forest Classifier | $58.9 \pm 0.2\%$ |
| Gradient Boosting Classifier | $58.9 \pm 0.2\%$ |

Figure 10: Accuracy on the distinction task for different adversarial learning models.

encryption of the input, evaluation of the function and discrete logarithm. Regarding encryption and evaluation, the main overhead comes from the exponentiations and pairings which are implemented in the crypto library charm [3]. In return, the discrete logarithm is very efficient thanks to the reduction of the weights amplitude detailed in Figure 4.1.

| Functional key generation | $94 \pm 5$ms | | Evaluation time | $2.97 \pm 0.07$s |
|---|---|---|---|---|
| Encryption time | $12.1 \pm 0.3$ s | | Discrete logarithms time | $24 \pm 9$ms |

Table 1: Average runtime for the FE scheme using a 2,7 GHz Intel Core i7 and 16GB of RAM.

Table 1 shows that encryption time is longer than evaluation time, but a single encryption can be used with several decryption keys $dk_{q_i}$ to perform multiple evaluation tasks.

## 6 Conclusion

We have shown that functional encryption can be used for practical applications where machine learning is used on sensitive data. We have raised awareness about the potential information leakage when not all the network is encrypted and have proposed semi-adversarial training as a solution to prevent targeted sensitive features from leaking for a vast family of adversaries.

However, it remains an open problem to provide privacy-preserving methods for all features except the public ones as they can be hard to identify in advance. On the cryptography side, extension of the range of functions supported in functional encryption would help increase provable data privacy, and adding the ability to hide the function evaluated would be of interest for sensitive neural networks.

## Acknowledgments

This work was supported in part by the European Community's Seventh Framework Programme (FP7/2007-2013 Grant Agreement no. 339563 – CryptoCloud), the European Community's Horizon 2020 Project FENTEC (Grant Agreement no. 780108), the Google PhD fellowship, and the French FUI ANBLIC Project.

## Footnotes

[1]Note that we only present a simplified scheme here. In particular, the actual encryption is randomized, which is necessary to achieve IND-CPA security.

[2]https://github.com/pytorch/examples/blob/master/mnist/main.py

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
