[Supplementary Material]

## A  Functional Encryption and crypto tools

### A.1  Formal definition of Functional Encryption

Functional encryption relies on a pair of keys like in public key encryption: a master secret key msk and a public key pk. The public key pk can be shared and is used to encrypt the data, while the master secret key msk is used to build functional decryption keys $\mathsf{dk}_f$ for $f \in \mathcal{F}$. A user having access to $c$ an encryption of $x$ with pk and to $\mathsf{dk}_f$ can learn $f(x)$ but can't learn anything else about $x$.

We give the definition of Functional Encryption, originally defined in [12, 32].

**Definition A.1 (Functional Encryption)** *A functional encryption* scheme FE *for a set of functions* $\mathcal{F} \subseteq \mathcal{X} \to \mathcal{Y}$ *is a tuple of PPT algorithms* FE = (SetUp, KeyGen, Enc, Dec) *defined as follows.*

SetUp$(1^\lambda, \mathcal{F})$ *takes as input a security parameter* $1^\lambda$, *the set of functions* $\mathcal{F}$, *and outputs a master secret key* msk *and a public key* pk.

KeyGen$(\mathsf{msk}, f)$ *takes as input the master secret key and a function* $f \in \mathcal{F}$, *and outputs a functional decryption key* $\mathsf{dk}_f$.

Enc$(\mathsf{pk}, x)$ *takes as input the public key* pk *and a message* $x \in \mathcal{X}$, *and outputs a ciphertext* ct.

Dec$(\mathsf{dk}_f, \mathsf{ct})$ *takes as input a functional decryption key* $\mathsf{dk}_f$ *and a ciphertext* ct, *and returns an output* $y \in \mathcal{Y} \cup \{\bot\}$, *where* $\bot$ *is a special rejection symbol.*

### A.2  IND-CPA security

With notations of Appendix A.1, for any stateful adversary $\mathcal{A}$ and any functional encryption scheme FE, we define the following advantage.

$$\mathsf{Adv}_{\mathcal{A}}^{\mathsf{FE}}(\lambda) := \Pr\left[\beta' = \beta : \begin{array}{l} (\mathsf{pk}, \mathsf{msk}) \leftarrow \mathsf{SetUp}(1^\lambda, \mathcal{F}) \\ (x_0, x_1) \leftarrow \mathcal{A}^{\mathsf{KeyGen}(\mathsf{msk}, \cdot)}(\mathsf{pk}) \\ \beta \xleftarrow{\$} \{0, 1\}, \mathsf{ct} \leftarrow \mathsf{Enc}(\mathsf{pk}, x_\beta) \\ \beta' \leftarrow \mathcal{A}^{\mathsf{KeyGen}(\mathsf{msk}, \cdot)}(\mathsf{ct}) \end{array}\right] - \frac{1}{2},$$

with the restriction that all queries $f$ that $\mathcal{A}$ makes to key generation algorithm KeyGen$(\mathsf{msk}, \cdot)$ must satisfy $f(x_0) = f(x_1)$.

We say FE is IND-CPA secure if for all PPT adversaries $\mathcal{A}$, $\mathsf{Adv}_{\mathcal{A}}^{\mathsf{FE}}(\lambda) = \mathrm{negl}(\lambda)^2$.

### A.3  Bilinear Groups

Our FE scheme uses bilinear (or *pairing*) groups, whose use in cryptography has been introduced by [11, 24]. More precisely, given $\lambda$ a security parameter, let $\mathbb{G}_1$ and $\mathbb{G}_2$ be two cyclic groups of prime order $p$ (for a $2\lambda$-bit prime $p$) and $g_1$ and $g_2$ their generators, respectively. The application $e : \mathbb{G}_1 \times \mathbb{G}_2 \to \mathbb{G}_T$ is a pairing if it is efficiently computable, non-degenerated, and bilinear: $e(g_1^\alpha, g_2^\beta) = e(g_1, g_2)^{\alpha\beta}$ for any $\alpha, \beta \in \mathbb{Z}_p$. Additionally, we define $g_T := e(g_1, g_2)$ which spans the group $\mathbb{G}_T$ of prime order $p$.

We will denote by GGen a probabilistic polynomial-time (PPT) algorithm that on input $1^\lambda$ returns a description $\mathcal{PG} = (\mathbb{G}_1, \mathbb{G}_2, p, g_1, g_2, e)$ of an asymmetric bilinear group. For convenience, given $s = 1, 2$ or $T$, $n \in \mathbb{N}$ and vectors $\vec{u} := (u_1 \ldots u_n) \in \mathbb{Z}_p^n, \vec{v} \in \mathbb{Z}_p^n$, we denote by $g_s^{\vec{u}} := (g_s^{u_1} \ldots g_s^{u_n}) \in \mathbb{G}_s^n$ and $e(g_1^{\vec{u}}, g_2^{\vec{v}}) = \prod_{i=1} e(g_1, g_2)^{u_i \cdot v_i} = e(g_1, g_2)^{\vec{u} \cdot \vec{v}} \in \mathbb{G}_T$, where $\vec{u} \cdot \vec{v}$ is the inner product, i.e. $\vec{u} \cdot \vec{v} := \sum_{i=1}^n u_i v_i$.

$$
\begin{array}{|l|}
\hline
\underline{\mathsf{SetUp}(1^\lambda, \mathcal{F}_{n,B_x,B_y,B_f}):} \\[4pt]
\overline{\mathcal{PG} := (\mathbb{G}_1, \mathbb{G}_2, p, g_1, g_2, e) \leftarrow \mathsf{GGen}(1^\lambda),\ \vec{s}, \vec{t} \xleftarrow{\$} \mathbb{Z}_p^n,\ \mathsf{msk} := (\vec{s}, \vec{t}),\ \mathsf{pk} := \left(\mathcal{PG}, g_1^{\vec{s}}, g_2^{\vec{t}}\right)} \\[4pt]
\mathsf{Return}\ (\mathsf{pk}, \mathsf{msk}). \\[10pt]
\underline{\mathsf{Enc}\left(\mathsf{pk}, (\vec{x}, \vec{y})\right):} \\[4pt]
\gamma \xleftarrow{\$} \mathbb{Z}_p,\ \mathbf{W} \xleftarrow{\$} \mathsf{GL}_2,\ \text{for all } i \in [n],\ \vec{a}_i := (\mathbf{W}^{-1})^\top \begin{pmatrix} x_i \\ \gamma s_i \end{pmatrix},\ \vec{b}_i := \mathbf{W} \begin{pmatrix} y_i \\ -t_i \end{pmatrix} \\[4pt]
\mathsf{Return}\ \mathsf{ct} := \left(g_1^\gamma, \{g_1^{\vec{a}_i}, g_2^{\vec{b}_i}\}_{i\in[n]}\right) \in \mathbb{G}_1 \times (\mathbb{G}_1^2 \times \mathbb{G}_2^2)^n \\[10pt]
\underline{\mathsf{KeyGen}(\mathsf{msk}, q):} \\[4pt]
\mathsf{Return}\ \mathsf{dk}_f := \left(g_2^{q(\vec{s},\vec{t})}, q\right) \in \mathbb{G}_2 \times \mathcal{F}_{n,B_x,B_y,B_q}. \\[10pt]
\underline{\mathsf{Dec}\left(\mathsf{pk}, \mathsf{ct} := \left(g_1^\gamma, \{g_1^{\vec{a}_i}, g_2^{\vec{b}_i}\}_{i\in[n]}\right), \mathsf{dk}_q := \left(g_2^{q(\vec{s},\vec{t})}, q\right)\right):} \\[4pt]
out := e(g_1^\gamma, g_2^{q(\vec{s},\vec{t})}) \cdot \prod_{i,j\in[n]} e\left(g_1^{\vec{a}_i}, g_2^{\vec{b}_j}\right)^{q_{i,j}} \\[4pt]
\mathsf{Return}\ \log(out) \in \mathbb{Z}. \\
\hline
\end{array}
$$

Figure 10: Our functional encryption scheme for quadratic polynomials.

# B Our Quadratic Functional Encryption Scheme

## B.1 Proofs of IND-CPA security ans correctness

**Proof of Security**

To prove security of our scheme, we use the Generic Bilinear Group Model, which captures the fact that no attacks can make use of the representation of group elements. For convenience, we use Maurer's model [30], where a third party implements the group and gives access to the adversary via handles, providing also equality checking. This is an alternative, but equivalent, formulation of the Generic Group Model, as originally introduced in [31, 37].

We prove security in two steps: first, we use a master theorem from [6] that relates the security in the Generic Bilinear Group model to a security in a symbolic model. Second, we prove security in the symbolic model. Let us now explain the symbolic model (the next paragraph is taken verbatim from [4]).

In the symbolic model, the third party does not implement an actual group, but keeps track of abstract expressions. For example, consider an experiment where values $x, y$ are sampled from $\mathbb{Z}_p$ and the adversary gets handles to $g^x$ and $g^y$. In the generic model, the third party will choose a group of order $p$, for example $(\mathbb{Z}_p, +)$, will sample values $x, y \leftarrow_R \mathbb{Z}_p$ and will give handles to $x$ and $y$. On the other hand, in the symbolic model the sampling won't be performed and the third party will output handles to $X$ and $Y$, where $X$ and $Y$ are abstract variables. Now, if the adversary asks for equality of the elements associated to the two handles, the answer will be negative in the symbolic model, since abstract variable $X$ is different from abstract variable $Y$, but there is a small chance the equality check succeeds in the generic model (only when the sampling of $x$ and $y$ coincides).

To apply the master theorem, we first need to change the distribution of the security game to ensure that the public key, challenge ciphertext, and functional decryption keys only contain group elements whose exponent is a polynomial evaluated on uniformly random values in $\mathbb{Z}_p$ (this is called polynomially induced distributions in [6, Definition 10], and previously in [10]). We show that this is possible with only a negligible statistical change in the distribution of the adversary view.

<sub>500</sub> After applying the master theorem from [6], we prove the security in the symbolic model (cf.
<sub>501</sub> Appendix D.1), which simply consists of checking that an algebraic condition on the scheme in
<sub>502</sub> satisfied.

<sub>503</sub> **Theorem B.1 (IND-CPA Security in the Generic Bilinear Group Model)** *For any PPT adver-*
<sub>504</sub> *sary $\mathcal{A}$ that performs at most $Q$ group operations against the functional encryption scheme described*
<sub>505</sub> *on 10, we have, in the generic bilinear group model:*

$$\mathsf{Adv}_{\mathcal{A}}^{\mathsf{FE}}(\lambda) \leq \frac{12 \cdot (6n + 3 + Q + Q')^2 + 1}{p},$$

<sub>506</sub> *where $Q'$ is the number of queries to* $\mathsf{KeyGen}(\mathsf{msk}, \cdot)$.

<sub>507</sub> The proof of this result is quite technical and can be found in the dedicated Appendix D.

<sub>508</sub> **Proof of Correctness**

For all $i, j \in [n]$, we have:
$$e(g_1^{\vec{a}_i}, g_2^{\vec{b}_j}) = g_T^{\vec{a}_i \cdot \vec{b}_j} = g_T^{x_i y_j - \gamma s_i t_j}$$

<sub>509</sub> since

$$\vec{a}_i \cdot \vec{b}_j = \left( (\mathbf{W}^{-1})^\top \begin{pmatrix} x_i \\ \gamma s_i \end{pmatrix} \right)^\top \cdot \left( \mathbf{W} \begin{pmatrix} y_j \\ -t_j \end{pmatrix} \right)$$

$$= \begin{pmatrix} x_i \\ \gamma s_i \end{pmatrix}^\top \mathbf{W}^{-1} \mathbf{W} \begin{pmatrix} y_j \\ -t_j \end{pmatrix} = x_i y_j - \gamma s_i t_j.$$

<sub>510</sub> Therefore we have:

$$out = e(g_1^\gamma, g_2^{q(\vec{s}, \vec{t})}) \cdot \prod_{i,j} e(g_1^{\vec{a}_i}, g_2^{\vec{b}_i})^{q_{i,j}} = g_T^{\gamma q(\vec{s}, \vec{t})} \cdot g_T^{\sum_{i,j} q_{i,j} x_i y_j - \gamma q_{i,j} s_i t_j}$$

$$= g_T^{\gamma q(\vec{s}, \vec{t})} \cdot g_T^{q(\vec{x}, \vec{y}) - \gamma q(\vec{s}, \vec{t})} = g_T^{q(\vec{x}, \vec{y})}.$$

<sub>511</sub> **Proof of Complexity**

<sub>512</sub> The complexity can be inferred from the decryption phase as detailed in Figure 10 and we compare
<sub>513</sub> this with previous quadratic FE schemes in Figure 11.

| FE scheme | $ct$ | $\mathsf{dk}_f$ | Dec | Assumption |
|---|---|---|---|---|
| [6, Sec. 3] | $\mathbb{G}_1^{6n+1} \times \mathbb{G}_2^{6n+1}$ | $\mathbb{G}_1 \times \mathbb{G}_2$ | $6n^2(E_1 + P) + 2P$ | SXDH, 3PDDH |
| [6, Sec. 4] | $\mathbb{G}_1^{2n+1} \times \mathbb{G}_2^{2n+1}$ | $\mathbb{G}_1^2$ | $3n^2(E_1 + P) + 2P$ | GGM |
| Ours | $\mathbb{G}_1^{2n+1} \times \mathbb{G}_2^{2n}$ | $\mathbb{G}_2$ | $2n^2(E_1 + P) + P$ | GGM |

Figure 11: Performance comparison of FE for quadratic polynomials. $E_1$ and $P$ denote exponentiation
in $\mathbb{G}_1$ and pairing evaluation, respectively. Decryption additionally requires solving a discrete
logarithm but this computational overhead is the same for all schemes and is therefore omitted here.

<sub>514</sub> **B.2 Detailed equivalence of the FE scheme with a neural network**

<sub>515</sub> **Proof of Linear Homomorphism**

<sub>516</sub> For all $(\vec{x}, \vec{y}) \in \mathbb{Z}_p^n \times \mathbb{Z}_p^n$, and $(\vec{u}, \vec{v}) \in \mathbb{Z}_p^n \times \mathbb{Z}_p^n$, given an encryption of $(\vec{x}, \vec{y})$ under the public
<sub>517</sub> key $\mathsf{pk} := (g_1^{\vec{s}}, g_2^{\vec{t}})$, one can efficiently compute an encryption of $(\vec{u}^\top \vec{x}, \vec{v}^\top \vec{y})$ under the public key
<sub>518</sub> $\mathsf{pk}' := (g_1^{\vec{u}^\top \vec{s}}, g_2^{\vec{v}^\top \vec{t}})$. Indeed, given

$$\mathsf{Enc}(\mathsf{pk}, (\vec{x}, \vec{y})) := (g_1^\gamma, \{g_1^{\vec{a}_i}, g_2^{\vec{b}_i}\}_{i \in [n]}),$$

<sub>519</sub> and $\vec{u}, \vec{v} \in \mathbb{Z}_p^n$, one can efficiently compute:

$$(g_1^\gamma, g_1^{\sum_{i \in [n]} u_i \cdot \vec{a}_i}, g_2^{\sum_{i \in [n]} v_i \cdot \vec{b}_i}),$$

which is $\mathsf{Enc}(\mathsf{pk}', (\vec{u}^\top \vec{x}, \vec{v}^\top \vec{y}))$, since:

$$\sum_{i\in[n]} u_i \cdot \vec{a}_i = \sum_{i\in[n]} u_i \cdot (\mathbf{W}^{-1})^\top \begin{pmatrix} x_i \\ \gamma s_i \end{pmatrix} = (\mathbf{W}^{-1})^\top \begin{pmatrix} \sum_{i\in[n]} u_i \cdot x_i \\ \gamma \sum_{i\in[n]} u_i \cdot s_i \end{pmatrix}$$

$$= (\mathbf{W}^{-1})^\top \begin{pmatrix} \vec{u}^\top \vec{x} \\ \gamma \vec{u}^\top \vec{s} \end{pmatrix}.$$

Similarly, we have:

$$\sum_{i\in[n]} v_i \cdot \vec{b}_i = \sum_{i\in[n]} v_i \cdot \mathbf{W} \begin{pmatrix} y_i \\ -t_i \end{pmatrix} = \mathbf{W} \begin{pmatrix} \vec{v}^\top \vec{y} \\ -\vec{v}^\top \vec{t} \end{pmatrix}.$$

## C   Additional results

### C.1   Influence of weight compression on the network performance

We show here that we can manage to compress significantly the network weights in order to have a very fast discrete logarithm without modifying the results and conclusions made throughout the article. The main and collateral model follow the same CNN structure as stated above, and the collateral accuracy is reported after 10 epochs of training.

| | |
|---|---|
| Main accuracy with compression | $97.72 \pm 0.30\ \%$ |
| Collateral accuracy with compression | $55.27 \pm 0.41\ \%$ |

Table 2: Impact of weight compression on the main and collateral accuracies

### C.2   Influence of alpha during adversarial training

To choose the best value for $\alpha$, we have chosen an output size of 4 which allows us to keep a very high main accuracy while reducing significantly the collateral one, as shown in Figure 4. We observe that the semi-adversarial training does not affect much the main accuracy for a large range of values for $\alpha$, while its impact on the collateral accuracy is decisive. Figure 12 illustrates the role of $\alpha$ and justify our choice of $\alpha = 1.7$. For this experiment, we have chosen for both networks a simple feed forward with a hidden layer of 32 neurons.

Figure 12: Trade-off between the main and collateral tasks accuracies as a function of $\alpha$

## D   Security proof of our FE scheme

*Proof.* For any experiment Exp, adversary $\mathcal{A}$, and security parameter $\lambda \in \mathbb{N}$, we use the notation: $\mathsf{Adv}_{\mathsf{Exp}}(\mathcal{A}) := \Pr[1 \leftarrow \mathsf{Exp}(1^\lambda, \mathcal{A})]$, where the probability is taken over the random coins of Exp and $\mathcal{A}$.

| $\underline{\mathsf{Exp}_1(1^\lambda, \mathcal{A})}:$ | $\underline{\mathsf{KeyGen}(\mathsf{msk}, f)}:$ |
|---|---|
| $(\mathbb{G}_1, \mathbb{G}_2, p, g_1, g_2, e) \leftarrow \mathsf{GGen}(1^\lambda),\ \vec{s}, \vec{t} \overset{\$}{\leftarrow} \mathbb{Z}_p^n$ | return $(g_2^{f(\vec{s},\vec{t})}, f)$. |
| $a, b, c, d \overset{\$}{\leftarrow} \mathbb{Z}_p$, set $\mathcal{PG} := (\mathbb{G}_1, \mathbb{G}_2, p, g_1^{ad-bc}, g_2, e)$ | |
| $\mathsf{msk} := (\vec{s}, \vec{t}),\ \mathsf{pk} := \left( \mathcal{PG}, g_1^{(ad-bc)\vec{s}}, g_2^{\vec{t}} \right)$ | |
| $\left( (\vec{x}^{(0)}, \vec{y}^{(0)}), (\vec{x}^{(1)}, \vec{y}^{(1)}) \right) \leftarrow \mathcal{A}^{\mathsf{KeyGen}(\mathsf{msk}, \cdot)}(\mathsf{pk})$ | |
| $\beta \overset{\$}{\leftarrow} \{0,1\},\ \gamma \overset{\$}{\leftarrow} \mathbb{Z}_p$ | |
| for all $i \in [n],\ \vec{a}_i := \begin{pmatrix} d & -c \\ -b & a \end{pmatrix} \begin{pmatrix} x_i^{(\beta)} \\ \gamma s_i \end{pmatrix},\ \vec{b}_i := \begin{pmatrix} a & b \\ c & d \end{pmatrix} \begin{pmatrix} y_i^{(\beta)} \\ -t_j \end{pmatrix}$ | |
| $ct =: \left( g_1^{\gamma(ad-bc)}, \{g_1^{\vec{a}_i}, g_2^{\vec{b}_i}\}_{i \in [n]} \right)$ | |
| $\beta' \leftarrow \mathcal{A}^{\mathsf{KeyGen}(\mathsf{msk}, \cdot)}(\mathsf{pk}, ct)$ | |
| Return 1 if $\beta' = \beta$ and for all queried $f$, $f(\vec{x}^{(0)}, \vec{y}^{(0)}) = f(\vec{x}^{(1)}, \vec{y}^{(1)})$. | |

Figure 13: Experiment $\mathsf{Exp}_1$, for the proof of Theorem B.1.

While we want to prove the security result in the real experiment $\mathsf{Exp}_0$, in which the adversary has to guess $\beta$, we slightly modify it into the hybrid experiment $\mathsf{Exp}_1$, described in 13: we write the matrix $\mathbf{W} \overset{\$}{\leftarrow} \mathsf{GL}_2$ used in the challenge ciphertext as $\mathbf{W} := \begin{pmatrix} a & b \\ c & d \end{pmatrix}$, chosen from the beginning. Then $\mathbf{W}^{-1} = \frac{1}{ad-bc} \begin{pmatrix} d & -b \\ -c & a \end{pmatrix}$.

The only difference with the IND-CPA security game as defined in Appendix A.2, is that we change the generator $g_1 \overset{\$}{\leftarrow} \mathbb{G}_1^*$ into $g_1^{ad-bc}$ for $a, b, c, d \overset{\$}{\leftarrow} \mathbb{Z}_p$, which only changes the distribution of the game by a statistical distance of at most $\frac{3}{p}$ (this is obtained by computing the probability that $ad - bc = 0$ when $a, b, c, d \overset{\$}{\leftarrow} \mathbb{Z}_p$). Thus,

$$\mathsf{Adv}_{\mathcal{A}}^{\mathsf{FE}}(\lambda) = \mathsf{Adv}_0(\mathcal{A}) \leq \mathsf{Adv}_1(\mathcal{A}) + \frac{3}{p}.$$

Note that in $\mathsf{Exp}_1$, the public key, the challenge ciphertext and the functional decryption keys only contain group elements whose exponents are polynomials evaluated on random inputs (as opposed to $g_1^{\mathbf{W}^{-1}}$, for instance). This is going to be helpful for the next step of the proof, which uses the generic bilinear group model.

Next, we make the generic bilinear group model assumption, which intuitively says that no PPT adversary can exploit the structure of the bilinear group to perform better attacks than generic adversaries. That is, we have (with $\mathsf{Exp}_2$ defined in 14):

$$\max_{\mathsf{PPT}\ \mathcal{A}} \left( \mathsf{Adv}_1(\mathcal{A}) \right) = \max_{\mathsf{PPT}\ \mathcal{A}} \left( \mathsf{Adv}_2(\mathcal{A}) \right).$$

In this experiment, we denote by $\emptyset$ the empty list, by $\mathsf{append}(L, x)$ the addition of an element $x$ to the list $L$, and for any $i \in \mathbb{N}$, we denote by $L[i]$ the $i$'th element of the list $L$ if it exists (lists are indexed from index 1 on), or $\bot$ otherwise.

Thus, it suffices to show that for any PPT adversary $\mathcal{A}$, $\mathsf{Adv}_2(\mathcal{A})$ is negligible in $\lambda$. The experiment $\mathsf{Exp}_2$ defined in Figure 14 falls into the general class of simple interactive decisional problems from [6, Definition 14]. Thus, we can use their master theorem [6, Theorem 7], which, for our particular case (setting the public key size $N := 2n + 2$, the key size $c = 1$, the ciphertext size $c^* := 4n + 1$, and degree $d = 6$ in [6, Theorem 7]) states that:

$$\mathsf{Adv}_2(\mathcal{A}) \leq \frac{12 \cdot (6n + 3 + Q + Q')^2}{p},$$

$$\begin{array}{|l|}
\hline
\underline{\mathsf{Exp}_2(1^\lambda, \mathcal{A}):} \\[4pt]
L_1 = L_2 = L_T := \emptyset,\ Q_{\mathsf{sk}} := \emptyset,\ \vec{s}, \vec{t} \xleftarrow{\$} \mathbb{Z}_p^n,\ a, b, c, d \xleftarrow{\$} \mathbb{Z}_p,\ \mathsf{append}(L_1, (ad - bc) \cdot \vec{s}), \\[2pt]
\mathsf{append}(L_2, \vec{t}),\ \beta \xleftarrow{\$} \{0, 1\} \\[2pt]
\left( (\vec{x}^{(0)}, \vec{y}^{(0)}), (\vec{x}^{(1)}, \vec{y}^{(1)}) \right) \leftarrow \mathcal{A}^{\mathcal{O}_{\mathsf{add}}, \mathcal{O}_{\mathsf{pair}}, \mathcal{O}_{\mathsf{sk}}, \mathcal{O}_{\mathsf{eq}}}(1^\lambda, p) \\[2pt]
\mathcal{O}_{\mathsf{chal}}\left( (\vec{x}^{(0)}, \vec{y}^{(0)}), (\vec{x}^{(1)}, \vec{y}^{(1)}) \right) \\[2pt]
\beta' \leftarrow \mathcal{A}^{\mathcal{O}_{\mathsf{add}}, \mathcal{O}_{\mathsf{pair}}, \mathcal{O}_{\mathsf{sk}}, \mathcal{O}_{\mathsf{eq}}}(1^\lambda, p) \\[2pt]
\text{If } \beta = \beta', \text{ and for all } f \in Q_{\mathsf{sk}},\ f(\vec{x}^{(0)}, \vec{y}^{(0)}) = f(\vec{x}^{(1)}, \vec{y}^{(1)}), \text{ output 1. Otherwise, output 0.} \\[8pt]
\underline{\mathcal{O}_{\mathsf{add}}(s \in \{1, 2, T\}, i, j \in \mathbb{N}):} \\[2pt]
\mathsf{append}(L_s, L_s[i] + L_s[j]). \\[8pt]
\underline{\mathcal{O}_{\mathsf{pair}}(i, j \in \mathbb{N}):} \\[2pt]
\mathsf{append}(L_T, L_1[i] \cdot L_2[j]). \\[8pt]
\underline{\mathcal{O}_{\mathsf{chal}}\left( (\vec{x}^{(0)}, \vec{y}^{(0)}), (\vec{x}^{(1)}, \vec{y}^{(1)}) \right):} \\[2pt]
\gamma \xleftarrow{\$} \mathbb{Z}_p,\ \mathsf{append}(L_1, \gamma(ad - bc)) \\[2pt]
\text{for all } i \in [n],\ \vec{a}_i := \begin{pmatrix} d & -c \\ -b & a \end{pmatrix} \begin{pmatrix} x_i^{(\beta)} \\ \gamma s_i \end{pmatrix},\ \mathsf{append}(L_1, \vec{a}_i),\ \vec{b}_i := \begin{pmatrix} a & b \\ c & d \end{pmatrix} \begin{pmatrix} y_i^{(\beta)} \\ -t_i \end{pmatrix}, \\[2pt]
\mathsf{append}(L_2, \vec{b}_i). \\[8pt]
\underline{\mathcal{O}_{\mathsf{sk}}(f \in \mathcal{F}_{n, B_x, B_y, B_f}):} \\[2pt]
\mathsf{append}(L_2, f(\vec{s}, \vec{t})),\ Q_{\mathsf{sk}} := Q_{\mathsf{sk}} \cup \{f\}. \\[8pt]
\underline{\mathcal{O}_{\mathsf{eq}}(s \in \{1, 2, T\}, i, j \in \mathbb{N}):} \\[2pt]
\text{Output 1 if } L_s[i] = L_s[j],\ 0 \text{ otherwise} \\
\hline
\end{array}$$

Figure 14: Experiment $\mathsf{Exp}_2$. Wlog. we assume no query contains indices $i, j \in \mathbb{N}$ that exceed the size of the involved lists.

where $Q'$ is the number of queries to $\mathcal{O}_{\mathsf{sk}}$, and $Q$ is the number of group operations, that is, the number of calls to oracles $\mathcal{O}_{\mathsf{add}}$ and $\mathcal{O}_{\mathsf{pair}}$, provided the following algebraic condition is satisfied:

$$\{\mathbf{M} \in \mathbb{Z}_p^{(3n+2)\times(3n+Q'+1)} : \mathsf{Eq}_0(\mathbf{M})\}$$
$$= \{\mathbf{M} \in \mathbb{Z}_p^{(3n+2)\times(3n+Q'+1)} : \mathsf{Eq}_1(\mathbf{M})\},$$

where for all $\mathbf{M}, b \in \{0, 1\}$,

$$\mathsf{Eq}_b(\mathbf{M}) : \begin{pmatrix} 1 \\ (AD - BC)\vec{S} \\ (AD - BC)\Gamma \\ D\vec{x}^{(b)} - \Gamma C\vec{S} \\ -B\vec{x}^{(b)} + \Gamma A\vec{S} \end{pmatrix}^\top \mathbf{M} \begin{pmatrix} 1 \\ \vec{T} \\ A\vec{y}^{(b)} - B\vec{T} \\ C\vec{y}^{(b)} - D\vec{T} \\ (f(\vec{S}, \vec{T}))_{f \in Q_{\mathsf{sk}}} \end{pmatrix} = 0,$$

where the equality is taken in the ring $\mathbb{Z}_p[\vec{S}, \vec{T}, A, B, C, D, \Gamma]$, and 0 denotes the zero polynomial. Intuitively, this condition captures the security at a symbolic level: it holds for schemes that are not trivially broken. The latter means that computing a linear combination in the exponents of target group elements that can be obtained from pk, the challenge ciphertext, and functional decryption keys, does not break the security of the scheme. We prove this condition is satisfied in D.1 below. $\square$

**Lemma D.1 (Symbolic Security)** *For any* $(\vec{x}^{(0)}, \vec{y}^{(0)}), (\vec{x}^{(1)}, \vec{y}^{(1)}) \in Z_p^{2n}$, *and any set* $Q_{\mathsf{sk}} \subseteq \mathcal{F}_{n,B_x,B_y,B_f}$ *such that for all* $f \in Q_{\mathsf{sk}}$, $f(\vec{x}^{(0)}, \vec{y}^{(0)}) = f(\vec{x}^{(1)}, \vec{y}^{(1)})$, *we have:*

$$\{\mathbf{M} \in \mathbb{Z}_p^{(3n+2)\times(3n+Q'+1)} : \mathsf{Eq}_0(\mathbf{M})\}$$
$$= \{\mathbf{M} \in \mathbb{Z}_p^{(3n+2)\times(3n+Q'+1)} : \mathsf{Eq}_1(\mathbf{M})\},$$

*where for all* $\mathbf{M}$, $b \in \{0,1\}$,

$$\mathsf{Eq}_b(\mathbf{M}) : \begin{pmatrix} 1 \\ (AD-BC)\vec{S} \\ (AD-BC)\Gamma \\ D\vec{x}^{(b)} - \Gamma C\vec{S} \\ -B\vec{x}^{(b)} + \Gamma A\vec{S} \end{pmatrix}^{\top} \mathbf{M} \begin{pmatrix} 1 \\ \vec{T} \\ A\vec{y}^{(b)} - B\vec{T} \\ C\vec{y}^{(b)} - D\vec{T} \\ (f(\vec{S},\vec{T}))_{f \in Q_{\mathsf{sk}}} \end{pmatrix} = 0,$$

*where the equality is taken in the ring* $\mathbb{Z}_p[\vec{S}, \vec{T}, A, B, C, D, \Gamma]$, *and* $0$ *denotes the zero polynomial.*

*Proof.* Let $b \in \{0,1\}$, and $\mathbf{M} \in \mathbb{Z}_p^{(3n+2)\times(3n+Q'+1)}$ that satisfies $\mathsf{Eq}_b(\mathbf{M})$. We prove it also satisfies $\mathsf{Eq}_{1-b}(\mathbf{M})$. To do so, we use the following rules:

**Rule 1** : for all $P, Q, R \in \mathbb{Z}_p[\vec{S}, \vec{T}, A, B, C, D, \Gamma]$, with $\deg(P) \geq 1$, if $P \cdot Q + R = 0$ and $R$ is not a multiple of $P$, then $Q = 0$ and $R = 0$.

**Rule 2** : for all $P \in \mathbb{Z}_p[\vec{S}, \vec{T}, A, B, C, D, \Gamma]$, any variable $X$ among the set $\{\vec{S}, \vec{T}, A, B, C, D, \Gamma\}$, and any $x \in \mathbb{Z}_p$, $P = 0$ implies $P(X := x) = 0$, where $P(X := x)$ denotes the polynomial $P$ evaluated on $X = x$.

Evaluating $\mathsf{Eq}_b(\mathbf{M})$ on $B = D = 0$ (using **Rule 2**), then using **Rule 1** on $P = C\Gamma S_i T_j$ for all $i, j \in [n]$, we obtain that:

$$\mathbf{M}_{n+2+i} \begin{pmatrix} 0 \\ \vec{T} \\ \mathbf{0} \\ \mathbf{0} \\ (f(\vec{S},\vec{T}))_{f \in Q_{\mathsf{sk}}} \end{pmatrix} = 0,$$

where $\mathbf{M}_{n+2+i}$ denotes the $n+2+i$'th row of $\mathbf{M}$.

Similarly, using **Rule 1** on $P = \Gamma A S_i T_j$ for all $i, j \in [n]$, we obtain that:

$$\mathbf{M}_{2n+2+i} \begin{pmatrix} 0 \\ \vec{T} \\ \mathbf{0} \\ \mathbf{0} \\ (f(\vec{S},\vec{T}))_{f \in Q_{\mathsf{sk}}} \end{pmatrix} = 0.$$

Thus, we have:

$$\forall \beta \in \{0,1\} : \begin{pmatrix} 0 \\ \mathbf{0} \\ 0 \\ D\vec{x}^{(\beta)} - \Gamma C\vec{S} \\ -B\vec{x}^{(\beta)} + \Gamma A\vec{S} \end{pmatrix}^{\top} \mathbf{M} \begin{pmatrix} 0 \\ \vec{T} \\ \mathbf{0} \\ \mathbf{0} \\ (f(\vec{S},\vec{T}))_{f \in Q_{\mathsf{sk}}} \end{pmatrix} = 0. \qquad (1)$$

Using **Rule 1** on $P = (AD-BC)S_iBT_j$ for all $i, j \in [n]$ in the equation $\mathsf{Eq}_b(\mathbf{M})$, we get that the coefficient $M_{i+1,n+1+j} = 0$ for all $i, j \in [n]$. Similarly, using **Rule 1** on $P = (AD-BC)S_iDT_j$ for all $i, j \in [n]$, we get $M_{i+1,2n+1+j} = 0$ for all $i, j \in [n]$. Then, using **Rule 1** on $P = (AD-BC)\Gamma BT_j$ for all $j \in [n]$, we get $M_{n+2,n+1+j} = 0$ for all $j \in [n]$. Finally, using **Rule 1** on $P = (AD-BC)\Gamma DT_j$ for all $j \in [n]$, we get $M_{n+2,2n+1+j} = 0$ for all $j \in [n]$. Overall, we obtain:

$$\forall \beta \in \{0,1\} : \begin{pmatrix} 0 \\ (AD-BC)\vec{S} \\ (AD-BC)\Gamma \\ \mathbf{0} \\ \mathbf{0} \end{pmatrix}^{\top} \mathbf{M} \begin{pmatrix} 0 \\ \mathbf{0} \\ A\vec{y}^{(\beta)} - B\vec{T} \\ C\vec{y}^{(\beta)} - D\vec{T} \\ \mathbf{0} \end{pmatrix} = 0. \qquad (2)$$

We write:

$$
\begin{pmatrix} 0 \\ \mathbf{0} \\ 0 \\ D\vec{x}^{(b)} - \Gamma C \vec{S} \\ -B\vec{x}^{(b)} + \Gamma A \vec{S} \end{pmatrix}^{\top} \mathbf{M} \begin{pmatrix} 0 \\ \mathbf{0} \\ A\vec{y}^{(b)} - B\vec{T} \\ C\vec{y}^{(b)} - D\vec{T} \\ \mathbf{0} \end{pmatrix}
$$

$$
= \sum_{i,j \in [n]} \begin{pmatrix} Dx_i^{(b)} - \Gamma C S_i \\ -B x_i^{(b)} + \Gamma A S_i \end{pmatrix}^{\top}
$$

$$
\times \left( m_{i,j}^{(1)} \begin{pmatrix} 1 & 0 \\ 0 & 1 \end{pmatrix} + m_{i,j}^{(2)} \begin{pmatrix} 1 & 0 \\ 0 & 0 \end{pmatrix} + m_{i,j}^{(3)} \begin{pmatrix} 0 & 0 \\ 1 & 0 \end{pmatrix} + m_{i,j}^{(4)} \begin{pmatrix} 0 & 1 \\ 0 & 0 \end{pmatrix} \right)
$$

$$
\times \begin{pmatrix} Ay_j^{(b)} - BT_j \\ Cy_j^{(b)} - DT_j \end{pmatrix}
$$

Evaluating the equation $\mathsf{Eq}_b(\mathbf{M})$ on $C = D = 0$ (by **Rule 2**), then using **Rule 1** on $P = \Gamma ABS_i T_j$ for all $i, j \in [n]$, we obtain $m_{i,j}^{(3)} = 0$ for all $i, j \in [n]$. Evaluating the equation $\mathsf{Eq}_b(\mathbf{M})$ on $A = B = 0$ (by **Rule 2**), then using **Rule 1** on $P = \Gamma CDS_i T_j$ for all $i, j \in [n]$, we obtain $m_{i,j}^{(4)} = 0$ for all $i, j \in [n]$. Evaluating the equation $\mathsf{Eq}_b(\mathbf{M})$ on $A = B = C = D = 1$ (using **Rule 2**), then using **Rule 1** on $P = \Gamma S_i T_j$ for all $i, j \in [n]$, using the fact that $m_{i,j}^{(3)} = m_{i,j}^{(4)} = 0$ and (1), we obtain $m_{i,j}^{(2)} = 0$ for all $i, j \in [n]$. Using **Rule 1** on $P = \Gamma(AD - BC)S_i T_j$ for all $i, j \in [n]$ in the equation $\mathsf{Eq}_b(\mathbf{M})$, we obtain that for all $i, j \in [n]$,

$$
m_{i,j}^{(1)} = \mathbf{M}_{n+2} \begin{pmatrix} 0 \\ \mathbf{0} \\ \mathbf{0} \\ \mathbf{0} \\ (f_{i,j})_{f \in Q_{\mathsf{sk}}} \end{pmatrix},
$$

where $\mathbf{M}_{n+2}$ is the $n + 2$'th row of $\mathbf{M}$.

Putting everything together, can write

$$
\begin{pmatrix} 0 \\ \mathbf{0} \\ 0 \\ D\vec{x}^{(b)} - \Gamma C \vec{S} \\ -B\vec{x}^{(b)} + \Gamma A \vec{S} \end{pmatrix}^{\top} \mathbf{M} \begin{pmatrix} 0 \\ \mathbf{0} \\ A\vec{y}^{(b)} - B\vec{T} \\ C\vec{y}^{(b)} - D\vec{T} \\ \mathbf{0} \end{pmatrix}
$$

as

$$
(AD - BC)\mathbf{M}_{n+2} \begin{pmatrix} 0 \\ \mathbf{0} \\ \mathbf{0} \\ \mathbf{0} \\ \left( f(\vec{x}^{(b)}, \vec{y}^{(b)}) - \Gamma f(\vec{s}, \vec{t}) \right)_{f \in Q_{\mathsf{sk}}} \end{pmatrix}
$$

$$
= (AD - BC)\mathbf{M}_{n+2} \begin{pmatrix} 0 \\ \mathbf{0} \\ \mathbf{0} \\ \mathbf{0} \\ \left( f(\vec{x}^{(1-b)}, \vec{y}^{(1-b)}) - \Gamma f(\vec{s}, \vec{t}) \right)_{f \in Q_{\mathsf{sk}}} \end{pmatrix}
$$

$$
= \begin{pmatrix} 0 \\ \mathbf{0} \\ 0 \\ D\vec{x}^{(1-b)} - \Gamma C \vec{S} \\ -B\vec{x}^{(1-b)} + \Gamma A \vec{S} \end{pmatrix}^{\top} \mathbf{M} \begin{pmatrix} 0 \\ \mathbf{0} \\ A\vec{y}^{(b)} - B\vec{T} \\ C\vec{y}^{(b)} - D\vec{T} \\ \mathbf{0} \end{pmatrix} \tag{3}
$$

595 where we use the fact that for all $f \in Q_{\mathsf{sk}}$, we have the equality $f(\vec{x}^{(b)}, \vec{y}^{(b)}) = f(\vec{x}^{(1-b)}, \vec{y}^{(1-b)})$.

596 Evaluating equation $\mathsf{Eq}_b(\mathbf{M})$ on $A = B = D = 0$ (by **Rule 2**), then using **Rule 1** on $\Gamma S_i C$ for all
597 $i \in [n]$, and using (1) and (3), we obtain that the coefficient $M_{n+2+i,1} = 0$ for all $i \in [n]$. Evaluating
598 $\mathsf{Eq}_b(\mathbf{M})$ on $B = C = D = 0$ (by **Rule 2**), then using **Rule 1** on $\Gamma S_i A$ for all $i \in [n]$, and using (1)
599 and (3), we obtain that the coefficient $M_{2n+2+i,1} = 0$ for all $i \in [n]$. Thus, we have:

$$\forall \beta \in \{0,1\} : \begin{pmatrix} 0 \\ \mathbf{0} \\ 0 \\ D\vec{x}^{(\beta)} - \Gamma C\vec{S} \\ -B\vec{x}^{(\beta)} + \Gamma A\vec{S} \end{pmatrix}^{\top} \mathbf{M} \begin{pmatrix} 1 \\ \mathbf{0} \\ \mathbf{0} \\ \mathbf{0} \\ \mathbf{0} \end{pmatrix} = 0. \tag{4}$$

600 Evaluating equation $\mathsf{Eq}_b(\mathbf{M})$ on $A = C = D = 0$ (by **Rule 2**), then using **Rule 1** on $BT_j$ for all
601 $i \in [n]$, and using (3), we obtain that the coefficient $M_{1,n+1+j} = 0$ for all $j \in [n]$. Evaluating
602 $\mathsf{Eq}_b(\mathbf{M})$ on $A = B = C = 0$ (by **Rule 2**), then using **Rule 1** on $DT_j$ for all $j \in [n]$, and using (3),
603 we obtain that the coefficient $M_{1,2n+1+j} = 0$ for all $j \in [n]$. Thus, we have:

$$\forall \beta \in \{0,1\} : \begin{pmatrix} 1 \\ \mathbf{0} \\ 0 \\ \mathbf{0} \\ \mathbf{0} \end{pmatrix}^{\top} \mathbf{M} \begin{pmatrix} 0 \\ \mathbf{0} \\ A\vec{y}^{(\beta)} - B\vec{T} \\ C\vec{y}^{(\beta)} - D\vec{T} \\ \mathbf{0} \end{pmatrix} = 0. \tag{5}$$

604 Overall, we have:

$$\mathsf{Eq}_b(\mathbf{M}) : \begin{pmatrix} 1 \\ (AD-BC)\vec{S} \\ (AD-BC)\Gamma \\ D\vec{x}^{(b)} - \Gamma C\vec{S} \\ -B\vec{x}^{(b)} + \Gamma A\vec{S} \end{pmatrix}^{\top} \mathbf{M} \begin{pmatrix} 1 \\ \vec{T} \\ A\vec{y}^{(b)} - B\vec{T} \\ C\vec{y}^{(b)} - D\vec{T} \\ (f(\vec{S},\vec{T}))_{f \in Q_{\mathsf{sk}}} \end{pmatrix} = 0$$

605 which implies the following relation, under (1),(2),(4),(5)

$$\begin{pmatrix} 1 \\ (AD-BC)\vec{S} \\ (AD-BC)\Gamma \\ \mathbf{0} \\ \mathbf{0} \end{pmatrix}^{\top} \mathbf{M} \begin{pmatrix} 1 \\ \vec{T} \\ \mathbf{0} \\ \mathbf{0} \\ (f(\vec{S},\vec{T}))_{f \in Q_{\mathsf{sk}}} \end{pmatrix}$$

$$+ \begin{pmatrix} 0 \\ \mathbf{0} \\ 0 \\ D\vec{x}^{(b)} - \Gamma C\vec{S} \\ -B\vec{x}^{(b)} + \Gamma A\vec{S} \end{pmatrix}^{\top} \mathbf{M} \begin{pmatrix} 0 \\ \mathbf{0} \\ A\vec{y}^{(b)} - B\vec{T} \\ C\vec{y}^{(b)} - D\vec{T} \\ \mathbf{0} \end{pmatrix} = 0$$

606 and then, under (3)

$$\begin{pmatrix} 1 \\ (AD-BC)\vec{S} \\ (AD-BC)\Gamma \\ \mathbf{0} \\ \mathbf{0} \end{pmatrix}^{\top} \mathbf{M} \begin{pmatrix} 1 \\ \vec{T} \\ \mathbf{0} \\ \mathbf{0} \\ (f(\vec{S},\vec{T}))_{f \in Q_{\mathsf{sk}}} \end{pmatrix}$$

$$+ \begin{pmatrix} 0 \\ \mathbf{0} \\ 0 \\ D\vec{x}^{(1-b)} - \Gamma C\vec{S} \\ -B\vec{x}^{(1-b)} + \Gamma A\vec{S} \end{pmatrix}^{\top} \mathbf{M} \begin{pmatrix} 0 \\ \mathbf{0} \\ A\vec{y}^{(1-b)} - B\vec{T} \\ C\vec{y}^{(1-b)} - D\vec{T} \\ \mathbf{0} \end{pmatrix} = 0.$$

607 Under (1),(2),(4),(5), this implies

$$\mathsf{Eq}_{1-b}(\mathbf{M}) : \begin{pmatrix} 1 \\ (AD - BC)\vec{S} \\ (AD - BC)\Gamma \\ D\vec{x}^{(1-b)} - \Gamma C\vec{S} \\ -B\vec{x}^{(1-b)} + \Gamma A\vec{S} \end{pmatrix}^{\top} \mathbf{M} \begin{pmatrix} 1 \\ \vec{T} \\ A\vec{y}^{(1-b)} - B\vec{T} \\ C\vec{y}^{(1-b)} - D\vec{T} \\ (f(\vec{S}, \vec{T}))_{f \in Q_{\mathrm{sk}}} \end{pmatrix} = 0$$

608 $\square$

## Footnotes

[2]In cryptography, the security parameter $\lambda$ is a measure of the probability with which an adversary can break the scheme. $\lambda$ or $1^\lambda$ means that the probability of breaking the scheme is $2^{-\lambda}$.