[Reviews · NeurIPS 2019]

Reviewer 1



The paper is very well written but I'm concerned whether the paper is a fit for NeurIPS. The main contribution to ML in Section 4 feels rather compressed, and by far most of the references got to venues in cryptography and information security (only one each for NeurIPS and ICML). Furthermore, given that the paper introduces a new functional encryption scheme, it would make to sense to submit to a venue where this receives due appreciation and scrutiny. Post rebuttal: I appreciate the author's efforts to introduce a new concept to the ML community, but I stand by my preference of scrutiny in a more appropriate venue.

Reviewer 2



The paper presents a novel FE scheme and discusses it in detail. I enjoyed reading it. However, the definition of this encryption scheme remains unclear in the end. The authors refer to Figure 10, which is not available. Without this crucial information, the remainder of the paper lacks clarity. However, the experiments are chosen carefully and highlight their finding. I wonder which important information the circles and lines in Figures 2 and 3 reveal to the reader? Eventually, it would have been more informative to replace these two Figures by Figure 10 from the appendix. Otherwise, the paper is not self-contained.

Reviewer 3



Summary of the work: This paper proposes a methodology to perform inference on encrypted data using functional evaluation. Authors develop a specific model consisting of private and public execution; the private (cypher-text) execution takes place a 2-layer perceptron with square activation functions in the hidden layer. The output of this 2-layer perceptron is revealed to the server, which runs another ML model to classify the input. Authors provide Functional Encryption tools to efficiently run the private part of the protocol. They also propose a strong points: - Authors clearly distinguish their work from other private inference scenarios: their target is applications where the client might not be "online" and cannot communicate in an SFE protocol. - The importance of off-line secure inference is well-motivated using examples. - The paper is well-written and different aspects on the work are explained effectively. - The scenario of having some labels to be revealed publicly (e.g., digit value) while keeping some private (e.g., font) is useful in many practical applications. weak points: - The threat model is not clearly explained. I would like to know the following: - It is not clear which party is the adversary; My assumption is that the server is untrusted. Given that, I cannot convince myself how the adversarial re-training step is meaningful: the server is the party who learns the private 2-layer perceptron using training data. From that point of view, the adversary itself has control over the underlying model and he can avoid adversarial re-training. - If the above scenario is not correct, it would be nice if the authors provide explanations to avoid confusion. Please specify which party trains each part of the ML model. - From the runtime results, the Encryption time (client's processing time) is around 4x the server's evaluation time. Therefore, outsourcing the evaluation to the server does not save much time (compared to the scenario that the client herself evaluates the model in plaintext). If the server is not willing to share the model parameters with the client, then we should assume that the server (the adversary) can train the model himself, which is makes the adversarial training step not so sensible. - Another major issue is the scalability of this approach, and whether it can be generalized to other datasets. The provided (private) model architecture is overly simple and is evaluated on MNIST-like data. I would like to see how the network architecture would perform (in terms of accuracy and runtime) for more complex datasets like CIFAR-10. - Minor: please describe what \theta in Section 4.2 represents.

[Author Response · NeurIPS 2019]

# Partially Encrypted Machine Learning using Functional Encryption

We graciously thank the reviewers for their helpful comments. We agree with the points made and will update the draft accordingly. We clarify some details of the article below.

**Relevance to NeurIPS.** As this venue primarily addresses an ML audience, we strive to make the cryptography exposition more self-contained. However, with the rise of privacy concerns in the ML community, we also find it useful to expose ML researchers to basic tools from cryptography to help popularize certain useful privacy-preserving techniques for data analysis.

**Response to Reviewer 2.** R2 mentions that functional encryption (FE) might not scale well and may echo R4 on this matter. This technique is still in its early stages, and while there exist FE schemes that are more flexible than ours, they are far too slow to be used in practice. In fact, this article shows that even if FE isn't as mature as homomorphic encryption or multi-party computation, we can already use it to propose concrete privacy-preserving techniques.

As our approach is focused on applications to ML and privacy, which are at the core of the article, we believe our contributions to be a good fit for NeurIPS. We do detail and reference many notions from cryptology. This is because the ML community may not be familiar with those new concepts, and we sought to introduce them carefully and rigorously. In return, classical notions of ML do not need to be referenced as much because they are well established. On the topic of the new FE scheme we introduce: it serves as a perfect pretext to explain the workings of FE schemes in general, but the merit of this scheme is more to accelerate computations than to bring theoretical advances in cryptology. Moreover, the use of adversarial ML to enhance privacy by reducing data leakage is, we believe, an interesting technique to avoid fully encrypted computations and bring efficient privacy-preserving techniques to the ML community.

**Response to Reviewer 3.** R3 points out that the encryption scheme is not clearly detailed, except for Figure 10 in the appendix. We agree that this scheme should be reintegrated into the core of the article, especially since it helps to understand how FE schemes work in practice. Figures 2 & 3 aim to illustrate how the private and public parts of the networks are organized (circles and lines represent neurons and connections between neurons respectively), and to explain where the adversary can try to extract knowledge. This information is already present in the text of the article so these figures could be removed to make room for Figure 10 and more detailed explanations.

**Response to Reviewer 4.** R4's main concern is that the threat model is not clearly explained. In particular, it is not clear where the adversary is and when the adversarial training phase takes place. To illustrate our explanation, we will use the spam filtering example. Data owners are the parties who exchange emails and they don't want to reveal sensitive data to the server, which is in charge of forwarding and processing the emails. The adversary is the server, which will try to gain access to private information. Adversarial training is done faithfully by the data owners to build a function $q$ so that a plain text evaluation $q(x)$ doesn't reveal private information about $x$ to the server. Once $q$ is built, a decryption key $\mathsf{dk}_q$ is provided to the server. The server may be malicious, but given $\mathsf{dk}_q$ it can only obtain $q(x)$ from an encryption $\mathsf{Enc}(x)$ and the choice of $q$ makes it really hard to recover private information about $x$ using $q(x)$. Figure 2 & 3 might also be confusing: during adversarial training we don't have a real adversary: we simulate a strong adversary and ensure that its inference power is negligible for private features. On the server at runtime, the adversary might behave differently but we obtain experimental guarantees for a large family of neural network attacks.

Another point raised by R4 is that encryption time is longer than evaluation time and, therefore, outsourcing computation might not be worthwhile. This is true for simple outsourcing scenarios, but in our context such as spam filtering, we can't trust the sender to perform the spam detection faithfully. As the recipient might not be online to do the filtering himself, we must use the intermediate server to perform this computation, but of course we don't want this server to read the emails' content. Last but not least, because of how FE works, one single encryption can be used with several decryption keys $\mathsf{dk}_{q_i}$ which means that the server could do several analyses: in addition to spam filtering, it could also detect if an email is urgent, contains abusive speech, etc. Note that, at encryption time, the functions for those analyses may not have been decided upon yet, so inference at that point may not have been an option.

With regard to R4's suggestion to explore other datasets, we fully agree and would have preferred to use more complex datasets. Besides the limited set of functions currently supported by functional encryption, our main concern is to find a dataset with two types of features and where the cross-distribution of features is balanced. To our knowledge, such a set of image data is not available but would be very beneficial to research into privacy-preserving ML.

Finally, in Section 4.2, $\theta$ stands for all the parameters of a neural network's section: $\theta_q$ for the privately-evaluated network, $\theta_{\mathsf{pub}}$ for the network predicting the public features and $\theta_{\mathsf{priv}}$ for the adversarial network predicting the private features.

[Meta-Review · NeurIPS 2019]

Privacy in machine learning is being studied by many in the community due to its importance in many practical applications. Most studies use Homomorphic Encryptions or Secure Multi Party Computation to achieve privacy. This work uses Functional Encryption (FE) which is a different set of tools with different capabilities. I find this a great contribution since it may influence future research by demonstrating another plausible direction. Moreover, the authors present a new FE scheme, tailored to work well with machine learning workloads. This triggers the question whether this paper should be evaluated by ML experts in NeurIPS or by privacy experts at Crypto for example. Publishing it in NeurIPS generates a problem, even during the review process since the NeurIPS community is not used to evaluated new cryptographic algorithms. Nevertheless, since the algorithm design makes adjustments to match ML, it is in the scope of NeurIPS to publish such paper.